# A Deeper Investigation of Drug Degradation Mixtures Using a Combination of MS and NMR Data: Application to Indapamide

**DOI:** 10.3390/molecules24091764

**Published:** 2019-05-07

**Authors:** Cécile Palaric, Roland Molinié, Dominique Cailleu, Jean-Xavier Fontaine, David Mathiron, François Mesnard, Yoann Gut, Tristan Renaud, Alain Petit, Serge Pilard

**Affiliations:** 1Plateforme-analytique, Institut de Chimie de Picardie FR CNRS 3085, Université de Picardie Jules Verne, 33 rue Saint Leu, 80039 Amiens, France; cecile.palaric@u-picardie.fr (C.P.); dominique.cailleu@u-picardie.fr (D.C.); david.mathiron@u-picardie.fr (D.M.); 2BIOPI EA 3900, Biologie des Plantes et Innovation, Université de Picardie Jules Verne, 1 rue des Louvels, 80000 Amiens, France; roland.molinie@u-picardie.fr (R.M.); jean-xavier.fontaine@u-picardie.fr (J.-X.F.); francois.mesnard@u-picardie.fr (F.M.); 3Technologie Servier, 27 rue Eugène Vignat, 45000 Orléans, France; yoann.gut@servier.com (Y.G.); tristan.renaud@servier.com (T.R.); alain.petit@servier.com (A.P.)

**Keywords:** forced degradations, indapamide, UPLC-UV-HDMS^E^, NMR, NUS, PLS

## Abstract

A global approach that is based on a combination of mass spectrometry (MS) and nuclear magnetic resonance (NMR) data has been developed for a complete and rapid understanding of drug degradation mixtures. We proposed a workflow based on a sample preparation protocol that is compatible to MS and NMR, the selection of the most appropriate experiments for each technique, and the implementation of prediction software and multivariable analysis method for a better interpretation and correlation of MS and NMR spectra. We have demonstrated the efficient quantification of the remaining active pharmaceutical ingredient (API). The unambiguous characterization of degradation products (DPs) was reached while using the potential of ion mobility-mass spectrometry (IM-MS) for fragment ions filtering (HDMS^E^) and the implementation of two-dimensional (2D) NMR experiments with the non-uniform sampling (NUS) method. We have demonstrated the potential of quantitative NMR (qNMR) for the estimation of low level DPs. Finally, in order to simultaneously monitor multi-samples, the contribution of partial least squares (PLS) regression was evaluated. Our methodology was tested on three indapamide forced degradation conditions (acidic, basic, and oxidative) and it could be easily transposed in the drug development field to assist in the interpretation of complex mixtures (stability studies, impurities profiling, and biotransformation screening).

## 1. Introduction

The stability testing of new drug substances and products guidelines Q1A(R2) that are issued by the International Conference on Harmonisation of Technical Requirements for Registration of Pharmaceuticals for Human Use (ICH) [1] suggests that stress studies should be carried out on an active pharmaceutical ingredient (API) to establish its inherent stability characteristics, such as the degradation pathways, which lead to the identification of degradation products (DPs) and hence support the suitability of the proposed analytical procedures. 

A lot of analytical techniques are used in order to assess the quality of drugs [2]. The most common method in the field of degradation product profiling is high-performance liquid chromatography (HPLC), coupled with ultraviolet/visible (UV/Vis) and/or mass spectrometric (MS) detectors. While UV/Visible is limited to the analysis of molecules bearing chromophores, MS, with the development of the electrospray ionization source (ESI), high resolution mass spectrometry (HRMS) instruments, tandem mass spectrometry (MS/MS) tools, and ultra-performance liquid chromatography (UPLC) systems has become the most accessible and powerful technique in obtaining structural information of small molecules in complex mixtures [3,4]. Nevertheless, MS is limited to the analysis of ionizable compounds and it requires specific calibration curves for quantification purposes. The use of nuclear magnetic resonance (NMR) spectroscopy is the other way to access unambiguous structural and quantification data [5,6]. However, this technique lacks sensitivity and suffers from signal overlapping when analyzing mixtures. Consequently, its application for drug degradation screening often needs a purification step [7] or the hyphenation with liquid chromatography (LC-NMR) [2,8]. New approaches that involve the combination of MS and NMR data have been recently developed for metabolomics studies [9,10] and food control [11]. They include specific sample preparation and dedicated experiments (MS/MS, two-dimensional (2D) NMR). 

The aim of this work was to transpose and adapt these new methodologies, which are mainly described for amino acid prediction/quantification, in the field of drug development. Our objective is to offer a global tool to the pharmaceutical industry for a deeper investigation of complex mixtures containing API and numerous other compounds. In a first approach, we focus on MS and NMR data combination for the investigation of degradation mixtures, for which the main expectations are: the quantitative determination of the remaining API, the structural elucidation of DPs structures for a better understanding of API degradation pathways, the estimation of DPs levels, and the untargeted elucidation of unexpected impurities. 

The main steps of our approach are based on an original sample preparation protocol, which leads to consistent MS and NMR information and the use of a pool of relevant MS and NMR experiments for qualitative and quantitative purposes. The degradation profiles of indapamide (4-chloro-*N*-(2-methyl-1-indoline-)-3-sulfamoylbenzamide), a well-known thiazide-like diuretic that is used in the treatment of hypertension [12], were studied in acidic, basic, and oxidative conditions to validate our workflow (Figure 1). Concerning this API and its DPs (Table 1), a recent and well documented article [13] reported the analytical methods that are commonly used for their identification in different matrices (pharmaceutical formulations, biological fluids). They are mainly based on HPLC-UV and LC-ESI-MS/MS techniques (LC abbreviation, refer to HPLC or to UPLC). However, NMR is never directly implemented on the crude degradation mixture, and the benefit of modern NMR experiments to assist and simplify MS data interpretation will be demonstrated hereafter. In addition, recent mass spectrometry technical improvements, such as the hyphenation of ion mobility separation (IMS) with UPLC-ESI-HRMS, have been evaluated in our work. Moreover, in recent years, the use of statistics in analytical chemistry [14] was reported to assist in the interpretation of pharmaceutical preparations [15]. In our study, we have investigated the benefit of partial least squares (PLS) regression to establish a rapid correlation between MS and NMR data set when several degradation conditions need to be simultaneously analyzed and processed. 

## 2. Results and Discussion

### 2.1. Methodology Development

The key steps of our workflow (Figure 1) include a sample preparation protocol, followed by LC-UV-MS acquisitions that were concurrently conducted with one-dimensional (1D) and 2D-NMR experiments. In the aim of MS and NMR data set combination, sample preparation was a critical point with important consequences for the accuracy of results. In our case, the main challenge was to obtain redundant information for both of the techniques. For MS, starting from the crude degradation reaction solutions, only a dilution step was necessary. Concerning NMR experiments, which were performed directly with the crude solutions, we have experimented a new buffer system: DMSO-*d6*/D_2_O (70:30, *v*/*v*) with 50 mM Tris buffer, which was found to be efficient both for sample solubilization and pH adjustment to avoid chemical shift variations and the quality of shimming (see Section 3.2). 

MS acquisitions rely on the hyphenation of ion mobility spectrometry (IMS) with UPLC-ESI-HRMS using an IMS-Q-time-of-flight (TOF) system. Over the last decade, time-of-flight (TOF) instruments have played an important role in MS for the high-sensitivity analysis of complex chromatographic separations [16]. In addition, the benefit of IMS dimension has recently been reported as a promising device for the screening of many compounds in complex matrices while using high-definition all-ion fragmentation (HDMS^E^) acquisitions [17]. This powerful acquisition mode was implemented in our workflow. This is a data independent analysis (DIA) method, alternatively including a low-energy (LE) scan for precursor ions identification and a high-energy (HE) scan for fragmentation. For the HE scan, IMS allows for the precise alignment of fragments ions with their precursor thanks to drift-time filtering without the need of dedicated MS/MS or MS^n^ experiments. Moreover, the development of IMS protocols in the pharmaceutical industry seems to be of growing interest [18]. Regarding the introduction of the sample in the ESI source, the privileged method when developing a combined MS-NMR approach is often the direct introduction mode, as illustrated in the publications of Bingol [9,10] and in our recent work on amino acid supplements [11]. However, this method is very dependent on ionization efficiency and it is greatly affected by ion suppression problems, due to the presence of salts in degradation media. That is why the LC-MS method, including a classical on-line UV detection, was preferred in the present work for the profiling of degradation reactions (see Section 3.3). For more information regarding bond connectivity and stereochemistry, and to confirm MS data, NMR predictions and analysis were required. The proposed MS structural hypotheses were predicted while using MNova software and ^1^H and 2D-NMR spectra were recorded. This was directly done on degradation media, avoiding the need of a time consuming purification step, following an experimental protocol that is based on our previous results [11]. ^1^H-NMR experiment was essential to obtain information regarding chemical shift and multiplicity. To confirm the atom space arrangement, 2D experiments, such as J-resolved (JRES) spectroscopy (coupling constant), heteronuclear single quantum correlation spectroscopy (HSQC) (^1^H-^13^C correlation), and total correlation spectroscopy (TOCSY) (^1^H-^1^H correlation) were performed. In addition to 2D-NMR, the non-uniform sampling (NUS) method was implemented to the pulse sequences to gain in resolution in the case of overlapping signals. For quantification purpose, the qNMR approach [6] using ^1^H-NMR spectra was privileged (see Section 3.3). Finally, a statistical approach was also implemented in order to examine multi-degradations at the same time (see Section 3.4).

The results obtained for the study of indapamide degradations are presented below, in accordance with strategy’s objectives.

### 2.2. Quantification of API in Degradation Mixtures

In general, 5% to 20% degradation of the drug substance have been considered as the target percentages for the validation of stability studies [19]. LC-UV-MS and NMR quantifications of indapamide were performed by calibration curves in the range 0 to 100% (see Section 3.2). They were characterized by good linearity (r^2^ > 0.99), which was evaluated five times by analyzing a set of six concentrations while using the different UV (275 nm), MS (ESI^+^ and ESI^-^), and ^1^H-NMR acquisitions. The limits of quantification (LOQ) for indapamide obtained by UV/MS and ^1^H-NMR are 0.003 and 0.4 mM, respectively. The limits of detection (LOD) and the LOQ values for each technique and the methods of calculation are presented in Appendix A (Appendix A). It should be noted that for LC-UV-MS no internal standard was added to the solutions to facilitate the transposition to drug predevelopment stage, where often no standards are available. 

Three reconstituted mixtures (M1/M2/M3, see Section 3.2), including API and two degradation products (DP1 and DP3, Table 1), were analyzed to validate our quantification method. The results are almost always in the range ± 5%, whatever the technique used, as exemplified in Figure 2 for M1 and in Appendix A for M2/M3 (Appendix A).

In order to assess whether the mean value of the five replicates contains the theoretical one of the concerned reconstituted mixture, tests of significance, using an interval of confidence at the confidence level of 95%, were successfully completed (labelled by *: Figure 2 for M1, Appendix A for M2 and M3). It should be pointed out that UV and NMR provided more true and accurate results comparatively to MS and they could be considered as the most robust techniques for quantification. The API quantification method was then successfully applied on the three degradation media (acidic: HCl, basic: NaOH, oxidative: Cu(II)), which led to coherent results between the different techniques, as shown for acidic hydrolysis in Figure 2 and in Appendix A (Appendix A) for the other conditions. It should be noted that, for these real mixtures, the API contents were unknown prior to quantification, preventing significance tests to be performed. 

Table 2 summarizes the respective % of remaining API, depending on the stress factor. Among the three conditions, only acidic hydrolysis gave a level of API degradation (~25%) in the range of the ICH recommendations. Under basic hydrolysis, indapamide showed a lower percentage of degradation than that reported in 2016 by Kaddah’s group [13], which is due to our less drastic conditions (0.1 N NaOH at 70 °C for 20 min versus 50 °C/75% RH for 14 days). The degradation of indapamide is complete regarding our oxidative conditions with Cu(II).

Consequently, the HCl degradation mixture will be used to illustrate the next steps of our MS-NMR combined approach.

### 2.3. Identification/Prediction of DPs Using HDMS^E^ Acquisitions and MNova Software

Our workflow was first designed to analyze each degradation condition as a single set of data, starting with LC-UV-MS results. The UV chromatogram that was obtained for acidic conditions is presented in Figure 3A and in Appendix A (Appendix A) for the other conditions. Besides the indapamide (API) peak located at 6.97 min, three other compounds were clearly observed at 3.22, 3.91, and 7.67 min, respectively. The high-energy (HE) scans of indapamide obtained without (Figure 3B) and with drift time filtering (Figure 3C) evidenced the advantage of the HDMS^E^ scanning for improving the structural elucidation (see more details in Section 3.3). Indeed, UNIFI software seeks the ions with the same drift time and eliminates the others (e.g., [M + Na]^+^ adduct at *m*/*z* 388.0493 and interference at *m*/*z* 419.9834) to obtain a clarified fragmented MS spectrum. In addition, starting from a known parent molecule (indapamide or referenced impurities), fragment ions structures are proposed. 

The API fragment ions predicted by UNIFI software in positive (Figure 3B,C) and negative ion modes (Appendix A, Appendix A) are conform to the indapamide structure [13]. The degradation products were also subjected to HDMS^E^ investigation to establish their fragment profiles and elucidate their structures. The peak eluting at 7.67 min showing a 2 Da mass difference when compared to API matches with the structure of dehydro-indapamide (C_16_H_14_N_3_O_3_SCl, DP1, impurity B) while using accurate mass measurements of parent and fragment ions in the positive (Figure 3D) and negative (Appendix A) ion modes. It should be noted that the position of the double bond on the indole moiety could be either intra-cyclic or carried by the methyl group, only NMR will be able to distinguish such isomers. Only the negative ion mode was available for unambiguous MS characterization for the t_R_ 3.22 min (Figure 3E). The presence of the chlorine isotope pattern, the determination of the elemental composition (C_7_H_6_NO_4_SCl), and the characteristic loss of CO_2_ match with 4-chloro-3-sulfamoylbenzoate structure (DP3). As for DP1, NMR will confirm the precise aromatic substituent group positions. 

The peak eluting at 3.91 min, a new impurity referenced as DP5, was only detected in the positive ion mode. An abundant [M + H]^+^ ion was observed at *m*/*z* 145.0766 and C_9_H_9_N_2_ was the only elemental composition that was proposed by UNIFI software. Two fragment ions at *m*/*z* 115.0546 (C_9_H_7_, –N_2_H_2_) and *m*/*z* 91.0547 (C_7_H_7_, –C_2_H_2_N_2_) were also generated under HE conditions (Figure 3F), which suggests a highly conjugated aromatic backbone. Interrogation of the Chemspider database led to the selection of seven structures that were compatible with MS results: 1H-benzo[c][1,2]diazepine (benzodiazepine), 3-amino-3-phenylacrylonitrile (beta-aminocinnamonitrile), 5-phenyl-1H-pyrazole, 4-phenyl-1H-pyrazole, 4-methylcinnoline, 3-methylcinnoline, and 1-methylphtalazine.

1D and HSQC spectra of the different hypotheses were predicted while using MNova software (V.12.0, Mestrelab Research S.L., Santiago de Compostela, Spain) and the different characteristics: coupling constant, multiplicity, intensity, and chemical shift are presented in Appendix A (Appendix A for DP1 and DP3, Appendix A for DP5).

Concerning the other degradation conditions, the LC-UV-MS chromatograms (Appendix A, Appendix A) mainly highlighted the presence of DP1 for the basic hydrolysis and the exclusive formation of this degradation product in oxidative condition. It could be interesting to note that, contrary to Kaddah’s group finding [13], 2-methyl-1H-indol-1-amine (DP2, Table 1) was not detected in our alkaline degradation profile, even after a targeted search of its protonated molecule.

### 2.4. Structures Validation and Quantification of Degradation Products Using NMR

Following our workflow (Figure 1), experimental ^1^H and ^13^C-NMR spectra were compared one by one against the predicted spectra of the MS hypothetic structures. Chemical shift, coupling constants, multiplicities, and integral values were examined. In this manner, it was possible to gather signals that belonged to the same molecule in order to eliminate structures that did not match the experimental spectra and so to determine the best candidate. To illustrate this methodology, the acidic degradation NMR spectra (^1^H, TOCSY, and HSQC) are presented in Figure 4. As mentioned above, in addition to API, three groups of signals are evidenced and labelled as 1, 3, and 5 on the 1D and 2D spectra, by analogy to DPs nomenclature.

DP1 has been confirmed as the impurity B of indapamide (4-chloro-*N*-[2-methyl-1H-indol-1yl]-3-sulfamoylbenzamide) with an intra-cyclic double bond. Indeed, the presence of a methyl group (δ 2.33 (d, *J* = 0.9 Hz, 3H_21_)) eliminated the possibility of an extra-cyclic double bond. For DP3, the correlations between H_1_, H_4_, H_6_, and their respective multiplicities showed that no modification occurs in the aromatic substitution of the benzoate moiety and confirmed the 4-chloro-3-sulfamoylbenzoate structure (prediction vs. experimental NMR assignments for DP1 and DP3 are presented in Appendix A, Appendix A). Similarly, the presence of DP1 was confirmed by NMR for indapamide alkaline and oxidative hydrolysis. 

For DP5, the predicted NMR spectra of the seven candidates that were suggested by MS were studied. To distinguish the correct structure, some characteristics were discriminating: (1) the presence of a methyl group (δ 2.92 (d, *J* = 0.4 Hz, 3H_11_)) in the acidic degradation ^1^H spectra has allowed for the elimination of four propositions (1H-benzo[c][1,2]diazepine, 5-phenyl-1H-pyrazole, 4-phenyl-1H-pyrazole, 3-amino-3-phenylacrylonitrile); (2) the existing coupling ^4^J between H of methyl and H of –CH eliminate 1-methylphtalazine; (3) the ^13^C chemical shift of the –CH located on the pyrazine cycle at 124.85 ppm on the experimental spectrum (Appendix A, Appendix A) was essential in distinguishing the two remaining structures and it has validated the final choice of 3-methylcinnoline (prediction at 121.95 ppm versus 145.49 ppm for 4-methylcinnoline, Appendix A, Appendix A) as the best candidate for DP5.

The complete assignment of DP1, DP3, and DP5 NMR signals in the acidic degradation mixture was validated using the 2D-HMBC experiment recorded at 900 MHz (Appendix A, Appendix A). It could be notified that DP5 (3-methylcinnoline) was never previously described for indapamide stability studies in acidic conditions, while 2-methyl-2,3-dihydro-1H-indole-1-amine was more commonly observed (Impurity C, DP4, Table 1, the standard eluted at t_R_ 2.99 min was not detected in our HCl degradation mixture). Moreover, it is well documented that the ring enlargement reaction of aminoindoles to cinnoline derivatives spontaneously occurs with the action of O_2_ under atmospheric pressure [20,21], which could explain the formation of DP5 during our sample preparation protocol involving a large number of sealed ampules.

The relevance of our combined approach for MS and NMR data correlation is clearly demonstrated for degradation product structural characterization, as illustrated for DP5 (Figure 5). Indeed, MS has allowed for the determination of the elemental composition (C_9_H_8_N_2_) of the molecule and of its two fragment ions obtained following a loss of N_2_H_2_ (*m*/*z* 115.0546) and of C_2_H_2_N_2_ (*m*/*z* 91.0547), respectively. These losses are in accordance with the chemical structure that was proposed by NMR, where an aromatic diazoethylene moiety, which involved the positions 1 (N), 2 (N), 3 (C), and 11 (C), is available for N_2_H_2_ and C_2_H_2_N_2_ eliminations. Concerning NMR, the assignment of ^1^H and ^13^C signals was in agreement with the 3-methylcinnoline structure. However, some deviation between the predicted chemical shifts and the experimental ones are observed. This is due to the fact that chemical shift in NMR depends on pH, ionic strength, and solvent, and the prediction software used for interpretation assistance generally did not take these parameters into account. That is why the benefit of a combined use of MS and NMR is clearly demonstrated to solve this kind of problem.

Our last challenge was to quantify the main indapamide DPs without the need for specific calibration curves. Unlike the API, DPs standards are often not available in the predevelopment stage of new drugs, excluding the use of techniques, such as UV or MS, for their quantification. In that way, the qNMR approach [6], which does not require calibration curves, was applied. This method based on the ratio measurement between the internal standard (Tris buffer) and impurity ^1^H signals offer an unbiased view of the sample composition, and the possibility of simultaneously quantifying multiple compounds in complex mixtures. The precision of measurements for DP1 and DP3 was established in terms of repeatability and reproducibility (Appendix A, Appendix A) using the three reconstituted mixtures (M1, M2, and M3). Table 2 reports the measured levels (>1%) for DP1, DP3, and DP5 in the different degradation media. 

### 2.5. Contribution of Pre-Processing Filter Approach Using UV/MS and NMR Data for Multi-Degradation Studies

For the simultaneous profiling of several stress testing conditions, a pre-processing filter approach could be considered. NMR and UV/MS spectra that were generated during stability studies contain many signals, whose identification and interpretation require significant effort. In this way, chemometrics could be a helpful tool in extracting, describing spectral differences, and in evaluating global changes in large set of samples. To achieve this, the application of programs, such as partial least squares (PLS) regression, has been used to search components to predict the Y values from a set of X variables [14,22]. The NMR (X) and UV/MS (Y) data set of forced degradation mixtures (HCl, NaOH, Cu(II)), with five replicates for each degradation condition, were employed in PLS model construction. Two dimensions, which explain 95% of the total variance of Y, were selected. Figure 6 shows the PLS results (score plot, regression line, and loadings) for DP1 (Indapamide, impurity B). This treatment allows for the discrimination of the three degradations (Figure 6A,B). According to PLS loadings plot, it was also possible to connect impurity NMR signals with a specific UV/MS response (Figure 6C). This methodology allowed for the grouping of NMR signals belonging to the same molecule faster and more efficiently than manual investigation, and thus facilitates the interpretation of NMR data. This strategy opens the way to high-throughput analysis of drug stability in the predevelopment stage.

## 3. Materials and Methods

### 3.1. Chemicals

ORIL Industrie (Bolbec, France) provided indapamide (API, 99.9%), 4-chloro-*N*-(2-methyl-1H-indol-1-yl)-3-sulfamoylbenzamide (DP1, 95.5%), 4-chloro-3-sulfamoylbenzoic acid (DP3, 95.9%), and 2-methyl-2,3-dihydro-1H-indol-1-amine (DP4, 95.6%). Hydrochloric acid, sodium hydroxide, copper (II) chloride, and Trizma^®^ base were purchased from Sigma-Aldrich (Saint-Quentin Fallavier, France). Deuterated reagents, such as deuterium oxide (D_2_O) (99.9% D), dimethylsulfoxide-*d6* (DMSO-*d6*) (99.9% D) and 3-(trimethylsilyl)-propionic acid-*d4*, sodium salt (TMSP) (98% D), were purchased from Euriso-top (St Aubin, France). Formic acid, acetonitrile, methanol, and water ULC/MS grade were purchased from Biosolve BV (Valkenswaard, The Netherlands). Leucine-enkephalin was used as the lock-mass standard, and the mass spectrometer calibration solution (Major Mix) was purchased from Waters (Manchester, UK).

### 3.2. Forced Degradation Protocols and Sample Preparation

#### 3.2.1. Forced Degradation Studies

Alkaline hydrolysis at 0.25 mg/mL was carried out by dissolving 25.0 mg of indapamide in 35.0 mL of CH_3_OH/CH_3_CN (50:50, *v*/*v*), and then 10.0 mL of 1 N NaOH was added. The volume was completed at 100.0 mL with water in a graduated flask. The solution was distributed in sealed ampules (19 ampules, each containing 5 mL of solution), which were placed in an oven heated at 70 °C for 20 min. 

Acid hydrolysis at 0.25 mg/mL was carried out by dissolving 25.0 mg of indapamide in 35.0 mL of CH_3_OH/CH_3_CN (50:50, *v*/*v*), and then 10.0 mL of 1 N HCl was added. The volume was completed at 100.0 mL with water in a graduated flask. As described for basic hydrolysis, the solution was distributed in sealed ampules, which were placed in an oven heated at 70 °C for 24 h. 

For each condition, the 19 ampules were pooled, 87.0 mL were recuperated and neutralized at pH 7 ± 0.5 in a 100.0 mL graduated flask while using either hydrochloric acid or/and sodium hydroxide, and the volume was completed with water. 

Oxidative degradation at 0.25 mg/mL was carried out by dissolving 25.0 mg of indapamide in 35.0 mL of CH_3_OH/CH_3_CN (50:50, *v*/*v*), and then 10.0 mL of 0.5 M CuCl_2_ was added. The graduated flask was completed at 100.0 mL with water. The adjustment of 87.0 mL of the oxidative degradation to pH 7 ± 0.5 was achieved in a 100.0 mL graduated flask using NaOH 10N and the volume was completed at 100.0 mL with water. In this condition, the copper salts were precipitated. This solution was centrifuged and the supernatant was retrieved.

#### 3.2.2. Indapamide Calibration Curve and Model Mixtures

Stock solutions of indapamide (API) and DP3 at a concentration of 0.683 mM and 10 mM, respectively, were prepared in 65% of H_2_O and 35% of CH_3_CN/CH_3_OH (50:50, *v*/*v*). The stock solution of DP1 (10 mM) was obtained in CH_3_CN/CH_3_OH (50:50, *v*/*v*). The pH of each stock solution was adjusted to 7 ± 0.5. Calibration ranges of indapamide at concentrations of 0.14, 0.27, 0.41, 0.55, and 0.68 mM in 65% of H_2_O and 35% of CH_3_CN/CH_3_OH (50:50, *v*/*v*) were prepared by dilution. Model mixtures were prepared from the stock solutions in the following proportions API/DP1/DP3 (%): M1 (90/5/1), M2 (80/10/5), and M3 (80/1/10).

#### 3.2.3. MS and NMR Samples

Indapamide calibration levels, model mixtures, and degradation solutions were used for the MS and NMR experiments. MS: 73 µL were dissolved in 927 µL of CH_3_CN/H_2_O (10:90, *v*/*v*) to achieve a maximum concentration level of 0.05 mM. NMR: samples were prepared by the evaporation of 10.25 mL of each solution at 30 °C under nitrogen flow while using a Stuart^®^ sample concentrator (Dominique Dutscher, Brumath, France) in order to reach 10 mM for the highest API concentration. Sonication for ten minutes helped to dissolve the dry matter in 0.7 mL of DMSO-*d6*/D_2_O containing: (1) Tris buffer (50 mM, pH = 7.0) to limit the chemical shift and as an internal standard for the precise and accurate quantitative measurement (^1^H-NMR signal for qNMR at 600 MHz, (Dimethyl sulfoxide-*d6*) δ 3.59 (s, 6H)); (2) trimethylsilyl-propionic acid sodium salt (TMSP-*d4*, 0.1 mg/mL) for chemical shift reference. Subsequently, 0.6 mL were transferred to a 5 mm NMR tube. Five replicates were prepared for both analytical techniques. The correspondence between MS and NMR concentrations for each level of API (%) is as follows: % of API/mM in MS/mM in NMR: 20/0.01/2, 40/0.02/4, 60/0.03/6, 80/0.04/8, and 100/0.05/10. 

Concerning the degradation studies, starting from neutralized solutions, only a dilution step was necessary for MS to reach a target concentration of 0.05 mM. For NMR, more parameters need to be considered: (1) the pH value has a strong impact on chemical shift accuracy and for similar compounds the inter-sample chemical shift variations are a recurring problem for structure assignment; (2) the solvent has to cover a large range of polarity to observe all of the components; (3) the lack of sensitivity of this technique requires a concentration step. To achieve these requirements, in several plant metabolomics studies, the mixture MeOD/D_2_O with phosphate buffer has often been described [23], while in metabonomic or biofluid studies, D_2_O with phosphate buffer is more frequently used [24]. However, DMSO-*d6* is often the first choice for pharmaceutical applications, because pharmaceutics molecules are often poorly water soluble. The phosphate buffer that was used in metabolomics that precipitated in DMSO-*d6* was replaced with Tris buffer. Consequently, we have chosen a new buffer system: DMSO-*d6*/D_2_O (70:30, *v*/*v*) with 50 mM Tris buffer [25]. Deuterated NMR samples were all controlled by LC-UV-MS to ensure that further degradation did not occur during the evaporation step.

A Sartorius ME235P (Goettingen, Germany) analytical balance was used to weigh the powders. The pH was controlled using a Mettler Toledo (Viroflay, France) FiveEasyPlus FEP20 benchtop pH meter.

### 3.3. MS and NMR Analysis

#### 3.3.1. LC-UV-MS Acquisition

The mass spectrometric behavior of all the compounds was studied in positive and negative ionization modes (ESI^+^/ESI^−^). UHPLC separations were performed while using an ACQUITY UPLC I-Class system coupled on-line with an ACQUITY PDA detector and the Vion IMS Q-TOF (Waters, Manchester, UK), which was equipped with an electrospray (ESI) ionization source (Z-spray) and an additional sprayer (Lock Spray) for the reference compound. The degradation products and API were separated while using an ACQUITY UPLC HSS T3 (100 × 2.1 mm, 1.8 µm) column from Waters, maintained at 45 °C. The elution was performed using a 0.5 mL/min mobile phase gradient of water (A) and acetonitrile (B), which both contained 0.1% formic acid, programmed, as follows (A:B): 90:10 (t = 0 min), 90:10 (t = 2 min), 50:50 (t = 8 min), 50:50 (t = 9 min), 20:80 (t = 10 min), 20:80 (t = 11 min), 90:10 (t = 12 min), and 90:10 (t = 15 min). One microliter of each sample was injected. The PDA detector was set to record wavelength from 190 to 400 nm, and the indapamide signal was processed at 275 nm for quantification. The ESI source was used a capillary voltage of 3 kV for positive mode (ESI^+^) and 2.5 kV for negative mode (ESI^−^) and the following conditions: cone voltage, 40 V; source offset, 40 V; source temperature, 120 °C; desolvation gas temperature, 450 °C; desolvation gas flow, 800 L/h; and, cone gas flow, 50 L/h. Nitrogen (>99.5%) was employed as the desolvation and cone gas. Mass calibration was carried out while using a Major Mix solution and Leu-enkephalin (*m*/*z* 556.2771) was used as the lock mass (50 µg/L in H_2_O/CH_3_CN/Formic acid 50:49.9:0.1, *v*/*v*/*v*) for accurate mass measurements. Data independent analysis (DIA) was performed in the high-definition MS^E^ (HDMS^E^) mode over the range m/z 50–1200 at 0.2 s/scan. Thus, two independent scans with different collision energies (CE) were alternatively acquired during the run: a low-energy (LE) scan at a fixed CE of 6 eV and a high-energy (HE) scan where CE was ramped from 20 to 40 eV in ESI^+^ and 10 to 15 eV in ESI^−^, respectively. The traveling-wave ion mobility separation (TWIMS) capabilities of the Vion were used for MS^E^ fragment ions alignment, while using instrument’s default settings (wave velocity: 250 m/s; wave height: 45 V) [17]. More precisely, the LE scans were used for the identification/quantification of API and DPs by the response of their respective [M + H]^+^ or/and [M − H]^−^ ions and the HE scans for their structural characterization. The HE scan takes advantage of the TWIM cell [26], which is located before the quadrupole time-of-flight mass spectrometer. Indeed, the parent and product ions exhibit the same drift time in the mobility cell allowing an alignment of the ions from a same molecule, which avoids false positives in the fragmentation pattern and simplifying the identification process. The time-of-flight (TOF) was operated in the sensitivity mode, which provides an average resolving power of approximately 50,000 (FWHM) in the scanning *m*/*z* range. The high resolution mass spectrometry (HRMS) data were recorded in the continuum mode. UNIFI software controlled the acquisition (V1.8, Waters).

#### 3.3.2. LC-UV-MS Processing

Data were evaluated with UNIFI software. Regarding MS and UV quantifications, data processing was automatically performed with Quantification Tof 2D and Quantify UV methods, respectively. For MS, the method used information regarding t_R_, accurate m/z and molecular formulae. Area values of the extracted ion chromatograms for protonated (ESI^+^) and deprotonated (ESI^−^) molecules were obtained from the LE scans, and these values were transferred to Excel (Microsoft Excel 2011 v. 14.3.9, Microsoft, Redmond, Washington, WA, U.S.). Linear regressions were performed with the LINEST Microsoft Excel function, while using five replicates for each concentration level of the standard solutions. For UV, a specific wavelength (275 nm), which corresponds to the maximum absorbance of indapamide, was chosen and the calculated area values were processed similarly to MS. Limits of detection (LOD) and quantification (LOQ) were determined using the formula: LOD or LOQ = k × Sa_0_/a_1_ with k equal at 3.3 for LOD and 10 for LOQ. Sa_0_ is the standard deviation of the intercept and a_1_ is the slope of linear equation (see Appendix A
Appendix A, [11]). The prediction of the indapamide DP structures was based on the fragmentation data that were extracted from the HE scans, followed by an evaluation with UNIFI software using databases (in-house and Chemspider) or compared with previously described fragmentation pathways [13].

#### 3.3.3. NMR Acquisition

All NMR spectra were recorded at 300 K on a Bruker Avance III 600 MHz spectrometer (600.13 MHz for proton frequencies; Wissembourg, France) equipped with a z-gradient inverse probe head (TXI, 5-mm tube). Before the acquisition, the tuning of the spectrometer was performed, as described in our previous work [11]. TOPSPIN (V3.2, Bruker) software was used to perform the acquisition and processing. The 1D spectra were acquired using the pulse sequence from the Bruker library *zgcppr*, with 32 scans of 128 K data points and spectral widths of 9009 Hz. T1 measurements were performed on calibration ranges while using an inversion recovery pulse sequence *t1ir* in order to fix the relaxation delay of ^1^H experiment (5 * T1). A 13 s relaxation delay was used. The following 2D-NMR experiments were used: 

2D-^1^H JRES (J-resolved) NMR spectra were obtained using the pulse sequence *jresqf* with a 4 s relaxation delay using four transients per 64 increments, which were collected into 64 K data points, using spectral widths of 9014 Hz along F2 and 50 Hz along F1.

Non-uniform sampling (NUS) based 2D-TOCSY (total correlation spectroscopy) spectra were acquired using the sequence from the Bruker library *dipsi2gpphz* [27], with eight scans per 4 K increments, which were collected into 4 K data points, while using spectral widths of 9009 Hz in both dimensions. A 4 s relaxation delay and a mixing time of 100 ms were used. The number of NUS sampling points was 256 complex points (6.25% sampling density of 4 K points). 

NUS based 2D-HSQC (heteronuclear single quantum correlation spectroscopy) spectra were acquired with the Bruker library *hsqcetgpprsisp2.2*, with a 2 s relaxation delay using 16 scans per 8 K increments, which were collected into 4 K data points, while using spectral widths of 9014 Hz in F2 and 26,412 Hz in F1. The number of NUS sampling points was 256 complex points (3.125% sampling density of 8 K points). 

The poisson-gap sampling schedule was used in TOCSY and HSQC experiments. For HSQC and TOCSY sensitivity and resolution enhancement, a non-uniform sampling (NUS) program was applied. In our case, NUS was not used to reduce the experimental time, but to increase the number of points in the indirect dimension, and thus mainly the spectral resolution. The NMR parameters were transposed and adapted from a recent publication [28].

#### 3.3.4. NMR Processing

All of the spectra were processed as described in the previous work [11]. For chemometric pretreatment, all 1D spectra, over the range of 9.0–0.5, were baseline-corrected with an intermediate correction, aligned according to least square method and binned in 209 buckets with intelligent bucketing using NMR Procflow software [29]. The quantification of indapamide and its DPs was performed on the 1D-NMR spectra. The peak zones of quantitative interest were labeled by * on NMR spectra and tables (Figure 4 and Appendix A). Integration zones of indapamide were reported on calibration ranges to define its concentration in the different samples, and the linear regressions were obtained, as described for MS. Absolute qNMR method using internal calibration (Trisbase) was performed to quantify the degradation products [6]. ^1^H and ^13^C spectra of the hypothetical structures that were proposed by MS were predicted using MNova software (V.12.0, Mestrelab Research S.L., Santiago de Compostela, Spain) to obtain information about: coupling constant, multiplicity, and chemical shift.

### 3.4. Statistical Analysis

#### 3.4.1. Quantification

Statistical analysis was performed using R [30]. A Student test at 5% was carried out to test the significance of the differences that were observed between samples and theory. The Relative Standard Deviation (%RSD), expressed as a percentage, was calculated from the ratio of the standard deviation to the mean.

#### 3.4.2. UV/MS and NMR Data Correlation

One matrix per data set (^1^H-NMR and UV/MS) was built and imported into SIMCA software (version 14.1, Umetrics, Umea, Sweden) for multivariate analysis, where each spectrum was pretreated and arranged in rows as a two-dimensional matrix per analytical device. Partial least squares (PLS1) is a bilinear model, where both predictor X, corresponding to NMR binning and dependent variable Y, corresponding to the signal t_R_/area in UV/MS are projected to a latent subspace that maximizes the covariance between them. PLS1 were implemented on NMR and UV/MS data in order to identify the specific degradation products [14,22]. Two parameters were calculated to evaluate model performance: the explained variance in the response vector Y matrix and the predictive capability of the model (Q^2^).

## 4. Conclusions

In the present work, a combined MS-NMR approach was designed to assist in the interpretation of drug stability studies and it was experimented on indapamide (API) using three reconstituted mixtures and three forced degradation solutions (HCl, NaOH, Cu(II)). Concerning the quantification of API, the results were always in the range ± 5%, whatever the technique used and without the need for an internal standard (IS) in LC-UV-MS experiments. We demonstrated the high efficiency of ion mobility filtering for MS (HDMS^E^) and the benefit of TOCSY/HSQC pulse sequences with NUS strategy for NMR for the structural characterization of degradation products (DPs). We also proposed an accurate and repeatable quantification method for degradation products using qNMR in order to follow the ICH recommendations. Finally, partial least squares (PLS) regression was evaluated to establish a clear correlation between UV/MS and NMR data and it has proved to be promising for analyzing several degradations at the same time. Other chemometric tools, such as the multiblock method, will be soon evaluated and implemented in our workflow.

## Figures and Tables

**Figure 1 molecules-24-01764-f001:**
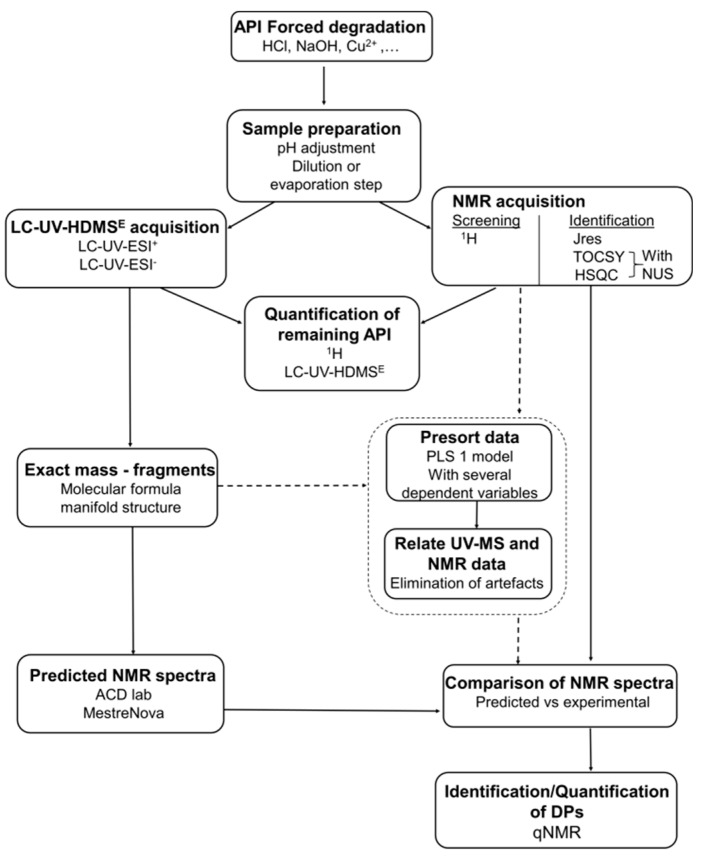
Workflow of the mass spectrometric-nuclear magnetic resonance (MS-NMR) approach.

**Figure 2 molecules-24-01764-f002:**
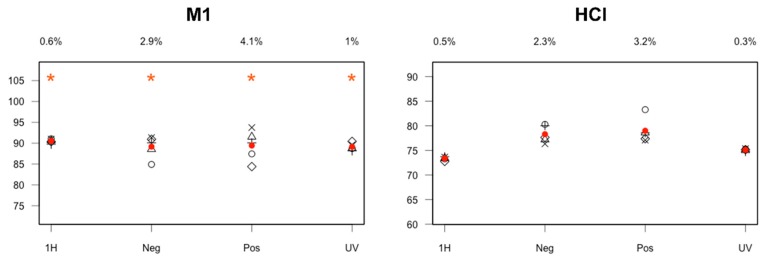
API quantification results for M1 and acidic degradation (%; Y axis) using the different analytical techniques (^1^H-NMR, ESI: negative and positive ion mode, UV at 275 nm; X axis) and for 5 replicates (⨯, ◊, ○, +, ∆ with ● representing the mean value). The coefficient of variation (CV) values (%) are specified at the top. * indicates that significance tests were successfully passed for reconstituted mixtures where theoretical values are available.

**Figure 3 molecules-24-01764-f003:**
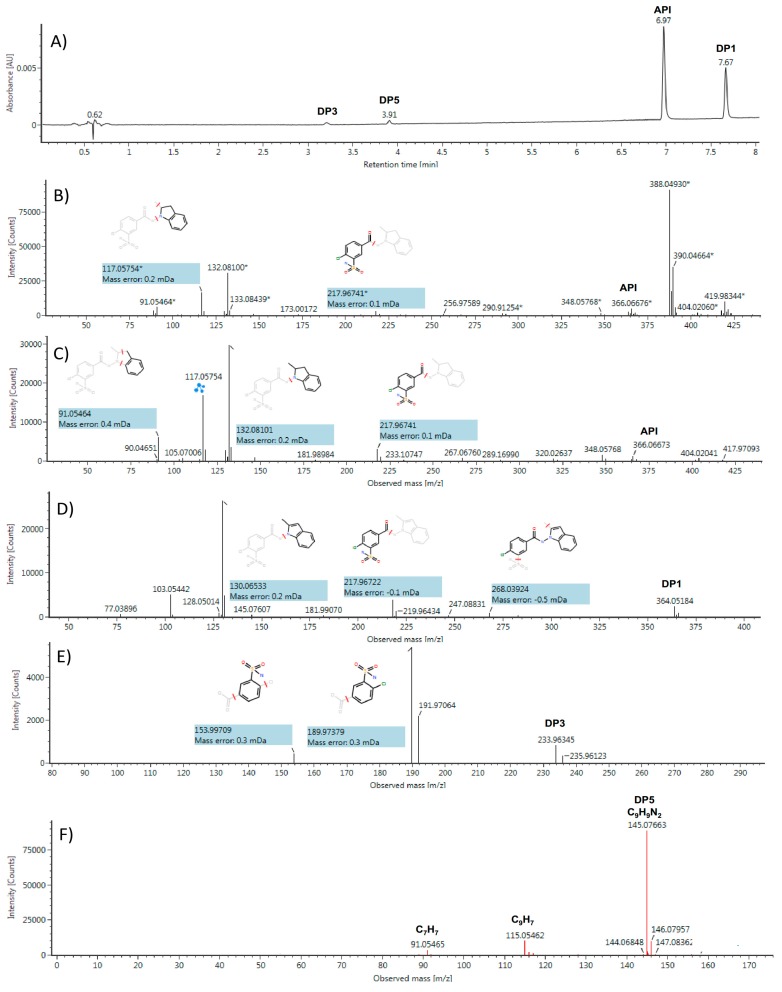
Profiling of indapamide (API) acidic hydrolysis using ultra-performance liquid chromatography/ultraviolet-ESI-high resolution mass spectrometry (UPLC/UV-ESI-HRMS): (**A**) UV chromatogram at 275 nm; (**B**) high-energy (HE) ESI^+^ spectrum of API without drift time filtering; (**C**) HE ESI^+^ spectrum of API with drift time filtering; (**D**) HE spectrum of DP1 in ESI^+^ with drift time filtering; (**E**) HE spectrum of DP3 in ESI^-^ with drift time filtering; and, (**F**) HE ESI^+^ spectrum of DP5 with drift time filtering.

**Figure 4 molecules-24-01764-f004:**
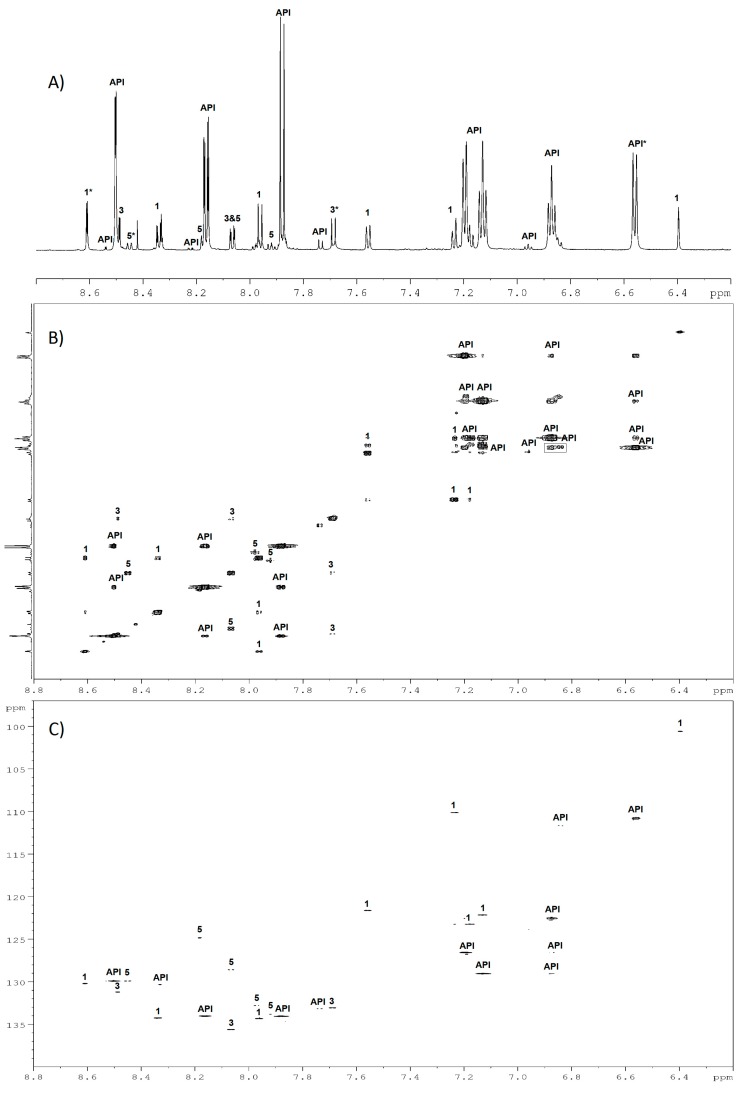
Identification of indapamide, DP1, DP3, and DP5 (labeled API, 1, 3, and 5) using one-dimensional (1D) and two-dimensional (2D)-NMR spectra of the acidic hydrolysis mixture: (**A**) ^1^H-NMR; (**B**) 2D-TOCSY; (**C**) 2D-heteronuclear single quantum correlation spectroscopy (HSQC) spectra. (*) corresponds to ^1^H selected for the quantification of API and its DPs.

**Figure 5 molecules-24-01764-f005:**
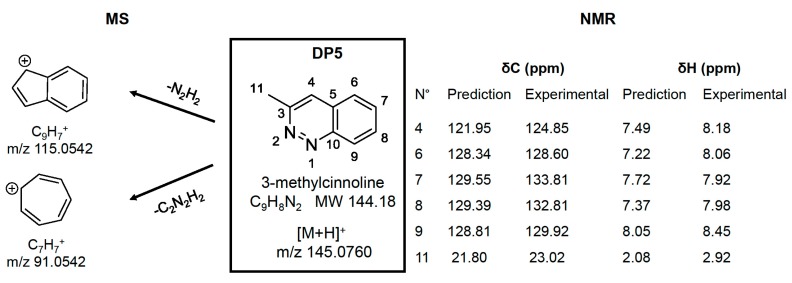
DP5 structural elucidation using the combined MS-NMR approach.

**Figure 6 molecules-24-01764-f006:**
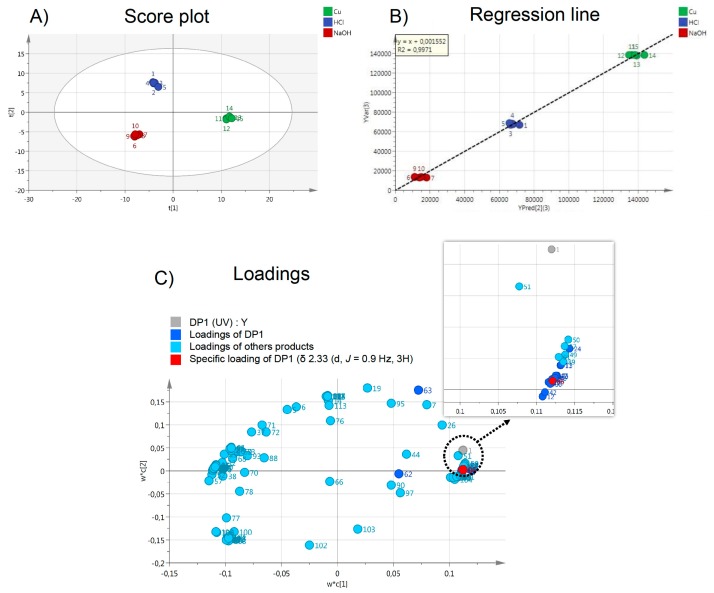
Partial least squares (PLS) results based on the DP1 signals observed in forced degradation mixtures (HCl, NaOH, Cu(II)). The NMR (X) and UV/MS (Y) data set were employed for PLS model construction, as follows: (**A**) score plot, (**B**) regression line, and (**C**) loadings. NB: loadings in dark blue indicate the chemical shift bins that present the most important correlation with the corresponding UV/MS response (as exemplified in red for the DP1 H_21_ characteristic proton, Table 1).

**Table 1 molecules-24-01764-t001:** Chemical structures, theoretical mass-to-charge ratios (*m*/*z*), ^1^H and ^13^C numbering of indapamide (API) and its main DPs (the nomenclature of the degradation products (DPs) was adapted from the publication of Kaddah’s group [13], 4-chloro-*N*-(2-methyl-1H-indol-1-yl)-3-sulfamoylbenzamide (DP1) and 2-methyl-2,3-dihydro-1H-indol-1-amine (DP4) correspond, respectively, to Imp B and Imp C of the European Pharmacopeia, 9th edition [12]). 2-methyl-1H-indol-1-amine (DP2) and DP4 are only characterized by electrospray ionization source (ESI^+^) and 4-chloro-3-sulfamoylbenzoate structure (DP3) by ESI^−^.

Compounds	Structure	Chemical Formula	[M + H]^+^ *m*/*z*	[M − H]^−^ *m*/*z*
Indapamide (API)	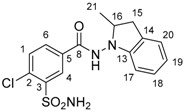	C_16_H_16_ClN_3_O_3_S	366.0674	364.0528
DP1	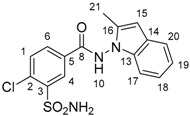	C_16_H_14_ClN_3_O_3_S	364.0517	362.0372
DP2	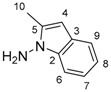	C_9_H_10_N_2_	147.0917	-
DP3	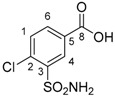	C_7_H_6_ClNO_4_S	-	233.9633
DP4	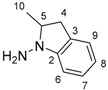	C_9_H_12_N_2_	149.1073 *	-

* weak (this compound mainly generates a [M − 2H + H]^+^ at *m*/*z* 147.0917).

**Table 2 molecules-24-01764-t002:** Summary of indapamide degradation behavior.

Stress Factor	Exposure	% of Remaining PA ± 2% (Mean of UV and ^1^H-NMR)	% of Individual DPs ± 5% (Based on qNMR)
Acid	0.1 N HCl at 70 °C for 24 h	74%	DP1: 14%; DP3: 8%; DP5: 5%
Base	0.1 N NaOH at 70 °C for 20 min	94%	DP1: 3%
Metal ion	0.05 M CuCl_2_, immediate	0%	DP1: 100%

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
