# Peer review of "A Deeper Investigation of Drug Degradation Mixtures Using a Combination of MS and NMR Data: Application to Indapamide"

_molecules, 2019, doi:10.3390/molecules24091764_

Round 1

Reviewer 1 Report

Chemicals: Add the purity of indapamide.

Sample preparation for MS and NMR: “Calibration ranges of indapamide at concentrations of 0”, 0 concentration?,  Concentration 0 is not correct.

Author Response

See Response to reviewer 1 in attached file.

Reviewer 2 Report

Reviewer’s comments

Date: 29 January 2019

Title: A deeper investigation of drug degradation mixtures using a combination of MS and NMR data:

application to indapamide.

Recommendations: Publish after minor corrections.

Comments

This is an interesting paper that describes how API could easily be analysed after drug degradation testing to quantify the amount of drug left and elucidate the various degradation products as required by ICH. Most techniques used LCMS only without using NMR spectroscopy which offers advantages as it does not rely on either chemical ionisation or presence of light absorbing chromophore in the case of UV based techniques.

Specific comments

1.                Figure 1 should be changed to a Table containing columns of compound names, structures, (M+H)+, (M-H)-, molecular formula for better view and clarity.

2.                2.4 MS and NMR data processing should be moved and combined with LC-UV-MS analysis line 152. Data processing follows LC-UV-MS analysis.

3.                NMR starting on line 219 should be moved NMR analysis (line 180); NMR processing follows NMR analysis.

4.                Both statistical analysis and multivariate analysis should be combined under one title ‘statistical analysis’.

5.                Some parts of the manuscript needed reorganising. For example from lines 246 to lines 259 need to be moved to the method section and not under results as what has been covered here are not results at all.

6.                Lines 264 to lines 319 should be moved to appropriate places in the method. Parts that have been described previously need not be repeated.

7.                Paragraphs that belong to the results should start from line 321.

8.                The legend for figure 7 should try to describe what the plots mean.

9.                Other comments and suggestions have been marked in the attached manuscript as ‘Marked-up’.

Author Response

See Response to reviewer 2 in attached file.

Reviewer 3 Report

Your approach is interesting, Your combination of techniques make sense but not completely the way you use it

It does not seem that You are acquiring the 1D NMR  spectra under quantitative conditions, Your relaxation delay of 13 s seems a bit short for a molecule the size as it also seems( zgcppr) that you use a 90 degree pulse. At 600 MHz i would expect emoting like 30 s and a 30 degreased pulse would do better (snd would not result in unfeasibly long experiment times) , but you could be OK, and could prove it by running an inversion recovery T1-determination. 

DP1, DP2,DP3, and DP4 are to be expected if You look at the structure. But You haven't addressed racemization at position 16, which is rather likely, but not directly detectable with Your chosen methods. Given that You are examining a drug, You should at least discuss this.

also, it seems (line 325, page 9 of 18) that You expect a linear calibration curve from Electro spray ionization, which is not generally the case. You evaluate linearity using r2, which is not sensitive enough to nonlinearity, but requires analysis of residuals, and subsequent fitting of  a linear and e.g. a quadratic calibration curve and analysis of the results.

use of PLS to identify the NMR and MS signals for the different species is OK, but actually makes the use of 2D Spectra slightly overkill, but nice that You have them

With proper buffering of the samples (AS YOU HAVE; NICE) You should be able to fit the "raw" NMR spectra dIrectly.

I am not happy with the way You determine LOD's and LOQ's. The ICH Q2 guideline section 6.4 recommends that You make a sample at the estimated limit and confirms the limit. " In cases where an estimated value for the detection limit is obtained by calculation or extrapolation, this estimate may subsequently be validated by the independent analysis of a suitable number of samples known to be near or prepared at the detection limit"

Author Response

See Response to reviewer 3 in attached file.

Reviewer 4 Report

The publication on the identification and quantification of impurities in a chosen active pharmaceutical ingredient focuses on the use of UHPLC-UV in combination with HRMS and NMR data.

The publication is well written and comprehensibly elaborated. However it will take serious re-iteration in terms of structure and presentation of the results. 

Most importantly, the results & dicussion part has a significant introduction/materials and methods part and first experimental results appear at section 3.3. A thorough restructuring is therefore needed.

Figure 2 orthography:

dependent variables

elimination of artefacts

comparison of NMR spectra

Figure 3 + S1: What does significance of the results mean? Why does HCl/NaOH treatment not have significance?

You use HRMS and 900 MHz NMR, please elaborate the structure of DP5 in more depth. Remove the list of possible compounds from Figure 4 f) it will not be readable. Rather propose a structure, which (in combination with NMR data) exlpains the fragments.

Figure 6: explain the considerable deviations of experimental vs predicted shifts!

Line by line comments:

21: ...industry, where a number...

26: replace tolerances by "deviations" or similar

30: DP (singular) levels

39: API means active PHARMACEUTICAL ingredient

40 leading to the identification

44 mass spectrometric detector

49 you also constructed calibration curves by diluting your mixes - please reformulate

58 amino acid (singular), drug stability (sing.)

59 offer a global tool ... to pharmaceutical industry

61 determination of the remaining api

68 multivariate tools were(?) considered

Materials and Methods:

If you did not use a certified reference material to construct calibration curves please at least indicate to purity / quality of your API and DP1-4.

Briefly also state how you defined LOD and LLOQ (not only in the supplement)

115 & 120 what does qs mean - please state

158 as follows

215 indapamide DP (sing.) structures

248 data, which 

257 DP (sing.) structures

258 the contribution of multivariate analysis was evaluated

259 which makes the use of a calibration curve unnecessary - I do not clearly understand how - you used a cal. curve...

275 APIs? pharmaceutical drugs?

Results and Discussion

3.3: R² is not a means to determine linearity it only describes deviations from a proposed function. Hence, either reformulate or perform a fitting test such as Mandel's! 

Renumber 2.4 to 3.4!

Discuss the results of drift time / no drift time filtering not only in a figure caption

381 - 391 You have all experimental data at hand. Be more concise on what you think the structure is - probably figure 4f/ figure 6 need to be updated

413 eliminated

426 This compound was not directly obtained

438 targeted search

471 loadings plot

Conclusion

Many of your claims are not to be found in the paper / not discussed thoroughly enough reduce to 3 or max. 5

494 of the API

500 and validated an accurate assay

Author Response

See Response to reviewer 4 in attached file.
